# Research

behaviour, ecology, environmental science

diving behaviour, water mass, continental shelf, Weddell seal, sex-specific variation, hidden Markov model

**Author for correspondence:**
Theoni Photopoulou
e-mail: theoni.photopoulou@gmail.com

# Sex-specific variation in the use of vertical habitat by a resident Antarctic top predator

Theoni Photopoulou[1,2], Karine Heerah[3], Jennifer Pohle[4] and Lars Boehme[1]

[1]Sea Mammal Research Unit, Scottish Oceans Institute, and [2]Centre for Research into Ecological and Environmental Modelling, University of St Andrews, St Andrews, UK
[3]Marine Bioacoustics Lab, Zoophysiology, Dept. Biology, Aarhus University, Aarhus, Denmark
[4]Department of Business Administration and Economics, Bielefeld University, Bielefeld, Germany

  TP, 0000-0001-9616-9940; JP, 0000-0003-2523-3030; LB, 0000-0003-3513-6816

Patterns of habitat use are commonly studied in horizontal space, but this does not capture the four-dimensional nature of ocean habitats (space, depth, and time). Deep-diving marine animals encounter varying oceanographic conditions, particularly at the poles, where there is strong seasonal variation in vertical ocean structuring. This dimension of space use is hidden if we only consider horizontal movement. To identify different diving behaviours and usage patterns of vertically distributed habitat, we use hidden Markov models fitted to telemetry data from an air-breathing top predator, the Weddell seal, in the Weddell Sea, Antarctica. We present evidence of overlapping use of high-density, continental shelf water masses by both sexes, as well as important differences in their preferences for oceanographic conditions. Males spend more time in the unique high-salinity shelf water masses found at depth, while females also venture off the continental shelf and visit warmer, shallower water masses. Both sexes exhibit a diurnal pattern in diving behaviour (deep in the day, shallow at night) that persists from austral autumn into winter. The differences in habitat use in this resident, sexually monomorphic Antarctic top predator suggest a different set of needs and constraints operating at the intraspecific level, not driven by body size.

## 1. Background

Understanding what parts of an ecosystem are important for species is a cornerstone of ecological research. Important habitat is often detected by proxy; if species regularly occur in a habitat, it must fulfil a life-history function. For large marine vertebrates, occurrence is usually measured using location data. However, identifying the drivers of marine population distributions from horizontal location data alone can be problematic for air-breathing deep-diving marine animals (e.g. [1]). This group spend most of their time underwater and are intrinsically difficult to observe. Depth is a fundamental dimension of their movement, and information is lost if dives are not considered. Vertical structuring of ocean habitats enhances productivity and creates predictable concentrations of resources [2,3]. Deep-divers target the increased prey density at steep physical gradients [4–6] and track its seasonality [7,8]. Seasonal variability in physical gradients is especially strong at the poles [9] due to seasonal extremes. For most deep-diving wide-ranging marine vertebrates, we lack a detailed understanding of the prey they consume and the structure and functioning of the ecosystems that support them.

For air-breathing divers like pinnipeds, seabirds, and cetaceans, dives are the result of the separation of two basic resources: air at the surface and prey at depth. Greater depths are more costly from a time-budget perspective, since transit likely excludes foraging [10], and they are physiologically more costly due to the metabolic requirements of hunting and digestion [11–13]. It follows that the habitat at these depth layers must be profitable in terms of

prey resources and that dives to regularly visited depth layers involve hunting and prey acquisition [10,14,15].

Understanding the environmental context of diving is key to linking behaviour to habitat and prey. Recent examples of integrative multi-species studies describing the spatial distribution of different marine animal guilds include [16–18]. However, surface variables do not reflect conditions at depth [19], and the methodology for relating diving behaviour to depth-varying environmental variables is underdeveloped in comparison. The collection of behavioural and *in situ* environmental data by satellite-linked animal-borne devices [20,21] is a way of filling this data-gap and allows us to relate depth-varying behaviour to depth-varying conditions. We posit that incorporating depth into habitat studies of diving animals (e.g. [22,23]) is critical for making good predictions of future distributions and detecting intraspecific variability that is hidden when taking a bird's-eye view of movement. We present an example of this in the Weddell seal (*Leptonychotes weddellii*), in its namesake shelf sea.

The southern Weddell Sea is a large embayment in the Southern Ocean's Atlantic sector. It is unique environmentally, due to physical processes that take place, leading to the formation of high-salinity water masses on the continental shelf [9,24,25]. Weddell seals occur all around the Antarctic coast and their diving behaviour has been studied since the earliest years of animal-borne instrument development [26,27]. Due to accessibility, most of what we know about their foraging ecology comes from East Antarctica. In these areas, seals seldom leave the continental shelf, where a much warmer and slightly fresher water mass (modified Circumpolar Deep Water) plays a central role [28,29]. In contrast, the southern Weddell Sea is only accessible by ship during austral summer. The interaction of Weddell seals with the hydrographically complex and varied Weddell Sea vertical habitat is not currently understood (but see [30–32]) and is likely to be different, owing to the availability of different water masses and deep ocean habitat. Previous work on foraging ecology in the Weddell Sea has found that dives target the surface, shallow epipelagic, and the mesopelagic region [30,33–35], while commuting through intermediate depths. This pioneering work has gone some way towards finding out what occurs at these depths, but knowledge of Weddell Sea shelf ecosystem remains poor.

Sex-specific differences in space use (horizontal and vertical) have been well-documented in air-breathing divers, but the underlying mechanisms remain unclear [36]. Many diving marine mammals are sexually dimorphic, making it difficult to disentangle size from other drivers (e.g. [37–39]). Being sexually monomorphic, Weddell seals present an opportunity to examine movement patterns independently from size. Several hypotheses exist for explaining sexual segregation in marine vertebrates, the main drivers being energy requirements and risk tolerance [36]. Different combinations of these may lead to divergent habitat use, time spent engaged in different behaviours, or risk exposure.

The high degree of serial correlation in animal movement metrics makes it necessary to use analytical approaches that account for it. Hidden Markov models (HMMs) [40,41] are time-series models, which allow us to learn about the underlying process from multivariate observations (multiple data streams). They have been used to model diving data from several marine predators and are powerful for making inferences about complex temporal patterns and responses to environmental features [42–46]. We use HMMs to identify Weddell

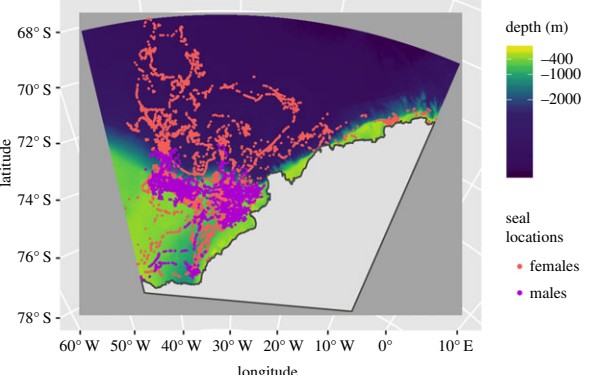

**Figure 1.** Satellite tracks from 19 instrumented Weddell seals carrying CTD-SRDL tags deployed in February 2011 (10 females and 9 males). The tracks are colour-coded by sex. The background colour represents bathymetry (depth in metres at a 0.5 km resolution). The colour bar is scaled so that light areas represent the continental shelf and dark areas represent the deep ocean. There is a deep trough on the continental shelf at 40° W, the Filchner Depression. These are predicted locations from a correlated random walk model fitted to the original data, accounting for the estimated CLS Argos location error, using the `foie-Gras` package in `R` [51]. (Online version colour.)

seal dive types or vertical movement states, and the effect of temporal covariates on the probability of switching between them. Geographical differences in diving activity [47] suggest that the two sexes may use different vertical habitat. We identify the depths where seals spend time hunting and link hunting activity to *in situ* oceanographic conditions, making this the first study to describe sex-specific preferences for vertical habitat in this circumpolar resident top predator. Although this is a single-species study, the approach can be used equally effectively for multi-species data.

## 2. Methods

### (a) Data collection
Movement and oceanographic data (salinity, temperature, and depth) were collected using telemetry instruments attached to the fur of adult Weddell seals in the southern Weddell Sea, Antarctica, after their annual moult. We used conductivity-temperature-depth satellite relay data loggers (CTD-SRDLs) [48,49], designed and manufactured by the Sea Mammal Research Unit Instrumentation Group (SMRU-IG), St Andrews, Scotland, UK. CTD-SRDLs employ the CLS Argos satellite system to relay data [50]. This produced a dataset from 19 Weddell seals (10 females, 9 males) instrumented in February 2011 (figure 1), during an oceanographic research program [9,52]. This is the largest single-year Weddell seal telemetry dataset from the Weddell Sea.

A detailed description of the deployment technique and individual deployment data are presented in [47,53]. Dive data were collected using algorithms developed for high-latitude, deep-diving seals [48,49,54] and CTD data were collected, calibrated, and processed according to MEOP (marine mammals exploring the oceans pole to pole) Consortium standards [20]. A summary of the data is presented in electronic supplementary material, S1.

### (b) Data processing
We consider a near-complete sequence of behaviour by combining the haulout, surface, and dive information returned by the tag (definitions: electronic supplementary material, S1). We do not consider short surface intervals between these events. We found that the amount of data reduces substantially after week 24 (13–19 June 2011) so we only consider data up to 19 June.

We calculated hunting depth and time spent hunting during dives using methods developed in [15]. They used high-resolution time-depth data and triaxial accelerometer data from Weddell seals to detect prey capture attempts (PrCAs) and found a strong correlation between vertical sinuosity, swimming speed, and the number of prey capture attempts. We use their estimated vertical speed threshold ($0.5\,m\,s^{-1}$) to extract segments of dives with low vertical speeds (hunting segments) and calculate the total duration of time spent in 'hunting' mode per dive [55]. We use the depth of the longest hunting segment, where most PrCAs occur, as the hunting depth in our analyses (electronic supplementary material, S2).

Only some dive profiles have a simultaneous CTD profile. To overcome this, we interpolated variables linearly in time, between time points where CTD data were available, and assigned them to the intervening dive times (electronic supplementary material, S3). We extracted bathymetry at the location of each seal dive after correcting the tracks by fitting a state-space model using the R package `foieGras` (electronic supplementary material, S3).

## (c) Statistical analysis

We fit a multivariate HMM to the haulout, surface, and dive data to classify diving behaviour and describe the relationship between dive types and two temporal covariates; time of day and season. The aim is to describe and classify diving behaviour, which includes foraging. The non-diving states in the model are known *a priori* from the data, so we only allow additional states to describe different diving behaviours. It was obvious from exploratory data analysis that female and male seals display different movement modes, which are differently affected by covariates, and therefore warrant separate models.

We use behavioural and environmental variables from each behavioural record (dive, surface, or haulout) to derive five data streams (state-dependent variables) which are modelled with the multivariate HMM: (1) behaviour duration (s), (2) hunting depth (m), (3) proportion of dive duration spent hunting (hunting time/duration: 0–1), (4) proportion of bathymetry reached at hunting depth (hunting depth/bathymetry: 0–1) (see electronic supplementary material, S3.3), and (5) salinity (psu) at hunting depth. These are the variables of interest that we wish to learn about and use to describe movement modes. Only behaviour duration is meaningful for non-dive behaviours, so this state-dependent variable alone is used to characterize haulout and surface events. All state-dependent variables are used to characterize dive states. We include the time series of salinity in the state process rather than as a covariate on the state transition probabilities (details: electronic supplementary material, S4).

Assuming contemporaneous conditional independence given the states, we model behaviour duration and hunting depth using a gamma distribution for non-zero positive values, proportion of dive time spent hunting as a beta distribution for values between zero and one, proportion of bathymetry reached as a single probability between zero and one, and salinity as a normal distribution. Salinity values are strictly positive but far from zero, so parameters need not be bounded.

Based on our data and the literature [26,32,56], we choose two covariates that act on the state transition probabilities: local time of day as a circular variable (cosine and sine), week of the year, and their interaction. The model formulation (the state-dependent variables, the temporal covariates, and the way the covariates were included in the model) was chosen carefully to make biological sense. The full model was used for both females and males. See electronic supplementary material, S4 for details on the model formulation and choice of covariates.

We estimate the parameters of the state-dependent distributions and the covariate effects using numerical maximization

of the likelihood, implemented in R [57], using the `nlm` function. The computation of the covariate-dependent transition probability matrices and the forward algorithm were coded in C++ (electronic supplementary material, S4). We used twenty sets of starting values for each model to ensure the algorithm found a global maximum (electronic supplementary material, S4). We use the Viterbi algorithm to calculate the most likely state sequence for each individual [41] (electronic supplementary material, S4). We checked the model fit by (1) computing the pseudo-residuals for each of the state-dependent variables, and by (2) simulating data from the model and checking how well it matched the observed data (electronic supplementary material, S5). There was no evidence of systematic lack of fit for either model.

# 3. Results

## (a) State-dependent distributions

The data from females and males were best described using different numbers of dive states, resulting in a five-state model for males and a six-state model for females (electronic supplementary material, S4). In each case, the two non-diving states were considered known and dive states were estimated from the data (females: 4 dive states, males: 3).

The estimated state-dependent distributions displayed in figure 2 show that female and male Weddell seals have three analogous vertical movement states. These are most easily thought of in terms of their hunting depth and proportion of bathymetry reached: (1) very short shallow dives, (2) slightly deeper dives in the epipelagic layer of the water column, and (3) deep dives with high probability of reaching the bottom (approx. 90%), where they access high salinity water masses found only on the continental shelf (table 1 and figure 2). Females also carry out a fourth dive type, in the pelagic layer of the water column. These dives are shorter than benthic dives and take place over the deep ocean, where the seals access lower salinity water masses (table 1 and figure 2). In males, pelagic dives appear to be absent, although their epipelagic dives are a bit deeper than in females. There was no compelling evidence of differences in proportion of time spent hunting between dive types or sexes.

The proportion of Viterbi decoded states differed between females and males (table 1). Both sexes are equally likely to haul out, and spend time at the surface, but males are almost twice as likely to carry out shallow dives (F: 17%, M: 31%) and benthic dives (F: 9%, M: 17%), while females are more likely to carry out epipelagic dives (F: 24%, M: 19%). Pelagic dives clearly make up a regularly used movement mode for females with 15% of records being estimated to belong to this state.

## (b) Covariate effects

The distribution of state occupancy (i.e. the equilibrium probabilities as functions of covariates) is shown in figure 3. These plots represent the probability (*y*-axis) of finding a seal in a given state (columns), during a week of the year (rows) throughout the 24-h cycle (*x*-axis). There is little diurnal variation in female or male shallow dives until winter, when they become more common at night. Female epipelagic dives happen predominantly at night in all weeks, and throughout the day in males. In both sexes, deep dives (pelagic and benthic) happen during the day, centred on local noon, (figure 3). This pattern is present throughout the seasons

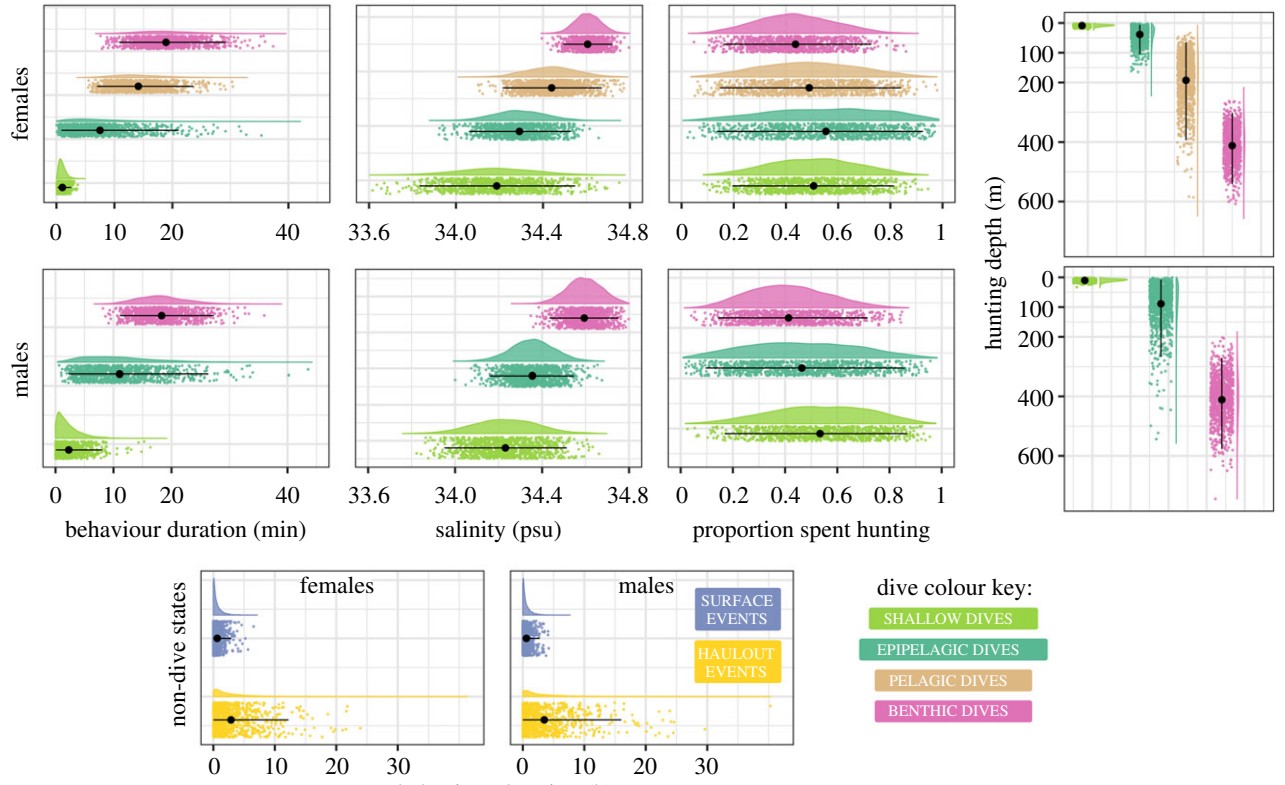

**Figure 2.** State density distributions for dive states (shallow, epipelagic, pelagic, and benthic dives) in female (top row) and male (middle row) Weddell seals. The state density distributions for non-diving states (haulout, surface) are shown for females (left) and males (right) in the bottom row. The density distributions are colour-coded by state, with a cloud of points below them. The cloud represents a realization of the estimated state-dependent distribution, based on a sample of 1000 points. The black point and line going through each coloured cloud of points represent the mean estimated by the model and the interval covering 95% of the probability mass. In females, all four dive states are present, while in males there are three dive states (no pelagic dives). (Online version colour.)

represented in our dataset, even when there is little light at local noon. We provide the state transition probabilities for the conditions shown in figure 3, in electronic supplementary material, S5.

A seasonal effect is evident in female diving behaviour, particularly pelagic and benthic diving. Early in the year (week 7) pelagic dives are common throughout the day, while benthic dives occur mainly in the day. By midwinter (week 24) deep dives are less common, replaced by night-time epipelagic diving. Male diving behaviour appears less seasonal. Daytime benthic dives occur throughout the observation period. Near midwinter they become less dominant and are complemented by shallow dives.

The difference in oceanographic regimes used by female and male seals is clear in the temperature-salinity diagrams associated with their dive profiles. Although temperature was not included in the model we can still visualize the Viterbi-decoded states in temperature-salinity (TS) space (figure 4). The distribution of states in TS space is unsurprising for the shallower states (shallow, epipelagic dives) because there is a limited range of available conditions in these surface layers. However, the deeper states show a trend: females use Modified Warm Deep Water and Warm Deep Water extensively for pelagic dives. These water masses are found along the diagonal line seen in the female TS diagram. By contrast, males in this dataset almost never visited areas with at-depth temperatures warmer than 0°C. The TS conditions associated with benthic dives are relatively consistent between females and males. In both plots, Ice Shelf Water and High Salinity Shelf Water, as well as its interface with precursors to Modified

Warm Deep Water, are visited regularly (water mass definitions: fig. 3 in [24]).

## 4. Discussion

We show that oceanographic data collected *in situ* by diving marine predators can be used to characterize their preferences for vertically distributed habitat and reveal intraspecific variability. We do this by modelling the relationship between diving and seasonal environmental conditions using HMMs. We find two movement strategies that in our dataset correspond to sex-specific differences. Females and males have some overlap in diving behaviour, but also important differences. Males mainly carry out benthic deep dives, consistent with their spatial location on the continental shelf, into winter (figure 1). By contrast, female deep dives are both benthic and pelagic, consistent with their extensive off-shelf autumn and winter distribution. Both sexes use high-salinity shelf water masses throughout, from February to June. Regular visits to this unique and inaccessible vertical habitat—the coldest possible liquid water habitat at −2°C—is worth the effort of carrying out deep dives, which are energetically costly [13]. Female diving behaviour follows a mixed seasonal pattern. This is in contrast with the comparatively aseasonal diving behaviour of males, suggesting that male diving behaviour does not track seasonally varying limitations or opportunities.

Dive types represent different vertical habitat types. We found that male seals avoid the pelagic zone almost entirely.

**Table 1.** Parameter estimates (mean, 95% confidence interval) for state-dependent distributions for female and male Weddell seals, from the respective HMMs—six states for females, five states for males. The proportion state occupancy is calculated as the proportion of observations labelled as each state. Note that there are four dive states for females to spread their activity over compared to three in males. This naturally leads to smaller proportions in the six-state scenario. However, if all states occur equally often, the proportions would all be smaller, which we do not see.

| | sex | haulout events | surface events | shallow dives | epipelagic dives | pelagic dives | benthic dives |
|---|---|---|---|---|---|---|---|
| duration | F | 2.8 h (2.7–3.0) | 37.9 min (36.4–39.5) | 1.0 min (59–64 s) | 7.4 min (7.2–7.7) | 14.1 min (13.9–14.2) | 19.0 min (18.7–19.2) |
| | M | 3.5 h (3.3–3.7) | 34.3 min (32.6–35.9) | 2.3 min (2.2–2.4) | 11.0 min (10.7–11.3) | — | 18.2 min (18.1–18.4) |
| depth (m) | F | — | — | 8.8 (8.6–8.9) | 38.5 (37.2–39.8) | 193.8 (189.0–198.6) | 413.1 (409.9–416.3) |
| | M | — | — | 10.2 (10.1–10.4) | 90.0 (86.8–93.2) | — | 411.8 (408.5–415.1) |
| proportion of dive spent hunting | F | | | 0.51 (0.50–0.51) | 0.55 (0.54–0.56) | 0.49 (0.48–0.50) | 0.43 (0.42–0.44) |
| | M | | | 0.53 (0.52–0.53) | 0.46 (0.46–0.47) | — | 0.42 (0.41–0.42) |
| proportion of bathymetry reached | F | | | 0.00 | 0.00 | 0.00 | 0.89 (0.72–1.00) |
| | M | | | 0.00 | 0.00 | — | 0.91 (0.77–1.00) |
| salinity at depth (psu) | F | | | 34.18 (34.178–34.192) | 34.29 (34.283–34.292) | 34.44 (34.433–34.444) | 34.61 (34.603–34.608) |
| | M | | | 34.23 (34.226–34.234) | 34.36 (34.351–34.360) | — | 34.59 (34.591–34.598) |
| state occupancy | F | 0.13 (0.10–0.17) | 0.21 (0.18–0.28) | 0.17 (0.12–0.23) | 0.24 (0.14–0.36) | 0.15 (0.07–0.30) | 0.09 (0.01–0.26) |
| | M | 0.13 (0.10–0.17) | 0.20 (0.12–0.26) | 0.31 (0.09–0.42) | 0.19 (0.12–0.31) | — | 0.17 (0.11–0.36) |

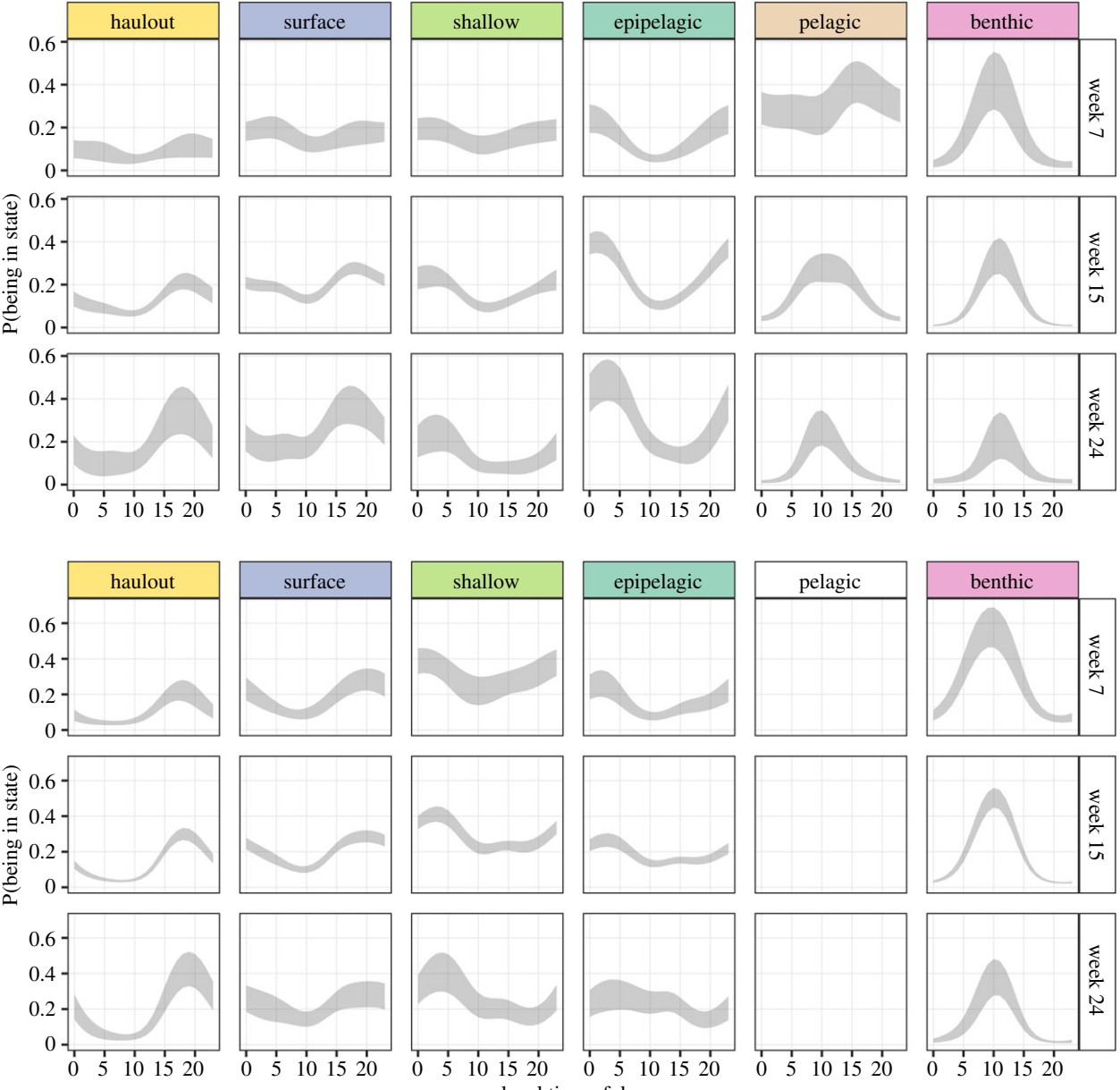

**Figure 3.** The probability (*y*-axis) of finding a seal in a given state (columns), during a week of the year (rows) throughout the 24-h cycle (*x*-axis). The grey ribbon represents the area enclosed by the lower and upper confidence bounds of the 95% interval of the estimated effect, which were calculated using the delta method. Time of day corresponds to local time calculated from the location of each data point. The female plots are shown in the top panel, and the males' in the bottom panel. Week 7 corresponds to late summer, week 15 to autumn, and week 24 to midwinter. (Online version colour.)

Dives down to the continental shelf benthos place seals in some of the coldest, densest water in the world. Harcourt *et al.* [58] found that males that dived deeply during the breeding season lost weight at a slower rate. This is evidence that deeper prey resources are likely to be particularly profitable, hinting at the productivity of on-shelf Antarctic habitats [59]. Possible reasons for leaving this profitable area could be intraspecific competition, density-dependent prey depletion, or pelagic resources being even more profitable. Females can move north to exploit them at shallower, less energetically costly depths, unhindered by the need to establish territories for the spring. This pattern is similar to southern elephant seals from Îles Kerguelen, where the males commonly stay in the sea ice and forage on the Antarctic continental shelf while females move north as winter progresses and forage in the marginal ice zone [60]. The life-history constraints that likely bring about this pattern for sexually dimorphic elephant seals are reversed for monomorphic Weddell seals—males are

limited by the need to establish territories, keeping them in the south, not females. In both cases, the marginal ice zone is clearly seasonally attractive for top predators. In light of the need for reliable sea ice habitat for breeding, and male territoriality, it seems likely that habitat segregation is based, at least partly, on females' lack of place-based constraints. Without knowing how profitable each of the two habitats is, we cannot say whether one sex excludes another or not. However, the increased predation risk from killer whales associated with lower sea ice concentrations, may point towards a high-risk high-rewards strategy in females [36].

The seasonality in female movement behaviour manifests latitudinally and vertically. In late summer, females are most likely to be found diving pelagically throughout the day, and benthically at midday, in contrast to what has been found in East Antarctica ([56] but also [28]). By midwinter, both deep dive types happen in the day. The hydrographic conditions associated with pelagic dives are compellingly clear:

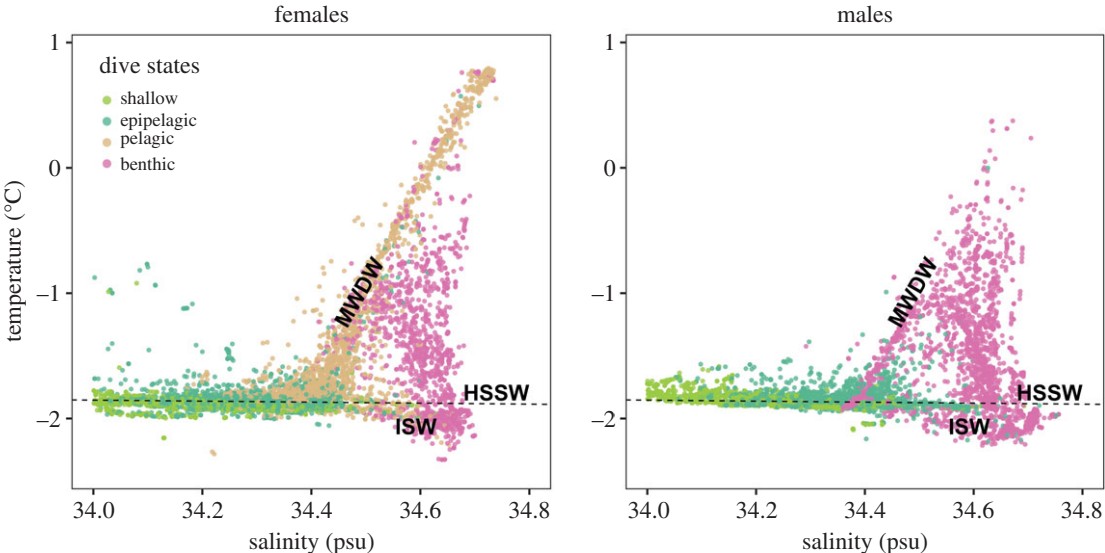

**Figure 4.** Temperature-salinity diagrams for dives from female and male seals coloured by Viterbi-decoded state. The points in each plot correspond to the conditions encountered by seals at the hunting depth of each dive. The water mass labels correspond to High Salinity Shelf Water (HSSW), Ice Shelf Water (ISW), and Modified Warm Deep Water (MWDW). These labels are transcribed from fig. 3 of Nicholls *et al.* [24]. (Online version colour.)

Modified Warm Deep Water, mixing with early Winter Water, and the continuum up the temperature gradient towards Warm Deep Water (fig. 4 and fig. 3 in Nicholls *et al.* [24] for reference). These conditions, like the marginal ice zone, have consistently been shown to be profitable to female phocid seals [4,23,28,61]. Sea ice has a high concentration of detritus and living organisms, which contribute to year-round biological productivity and community development, particularly under older ice floes [62,63]. This leads to a cascade of productivity, perhaps including the prey base exploited during the two shallow dive types we found. The existence of multi-year sea ice in the Weddell Sea may be the critical factor that allows female seals to forage successfully (and haul out) away from the coast, over deep water. The post-moult period finds female Weddell seals in need of replenishing their energy reserves after a period of reduced foraging due to lactation and the annual moult. A shallower pelagic diving strategy could afford the added energetic benefit of minimizing dive costs.

The lack of seasonality in male diving behaviour raises questions about the prey base they are exploiting. The end of summer sees the beginning of a reduction in the inflow of warm water onto the continental shelf in the southern Weddell Sea [52]. Warm water enters the shelf as Modified Warm Deep Water in the region of the Filchner Depression, especially the eastern and western edges of the Filchner Sill (figs 6 and 7 in [24], and also [52]). These physical changes must have some ecosystem effects but do not induce a discernible change in diving behaviour. However, a reduction in the probability of benthic dives and increase in epipelagic dives hint towards a shift in reliance towards surface layers in winter, despite the reduction in light availability. Changes not detectable by our approach include benthic prey type or prey behaviour. The TS diagram for males shows that some benthic dives occur in Modified Warm Deep Water (diagonal line starting at 34.4 psu in figure 4) but the clear majority of them occur in the denser, saltier water characteristic of sub-ice-shelf circulation and melting [64].

The shelf ecosystem that exists under these extreme temperature conditions is not well studied, but is likely to be very stable due to the narrow range of temperature and salinity conditions that can exist there [64]. It is known to support a large amount of biomass through Weddell seals and their likely prey, Antarctic toothfish (*Dissostichus mawsoni*) and other nototen fish (Antarctic silverfish *Pleurogramma antarcticum*, scaly rockcod *Trematomus loennbergii*). Stability and high biomass are attributes of mature ecosystems where large body size, narrow niche specialization and long, complex life cycles are observed [65]. These attributes fit with what we know about the Weddell Sea continental shelf from a top predator perspective. Riotte-Lambert and Matthiopoulos [66] show that learning and memory are emergent properties of animals in a system with moderate-to-high environmental predictability. At slightly lower predictability, more social information use is favoured. Although the degree of social information exchange is not known for Weddell seals, they have a substantial underwater vocal repertoire known to function socially, at least in the breeding season. This would place the predictability of their environment on the mid-to-upper end of the scale (fig. 1 in [66]) at a timescale comparable with the lifetime of a Weddell seal. As non-migratory top predators, Weddell seals are likely to be instrumental in the stability of the Antarctic shelf ecosystem, forming a feedback loop between movement ecology and environmental predictability [66].

Historically, bottom trawls found high nototen biomass at 600 m on the southern slopes of the Filchner Depression (especially *D. mawsoni*, *P. antarcticum*, and *T. loennbergii*) [59]. This is in contrast with the narrow eastern shelf where overall fish biomass was lower (fig. 3 in Ekau [59]), and which seals in our dataset tend to avoid. Over the Filchner Depression Antarctic toothfish were only caught in summer at 420–670 m depth and dominated in terms of biomass, along with scaly rockcod and large specimens of Antarctic silverfish. The distribution and temporal availability of these high-biomass, large-bodied fish is highly consistent with the location and dive depth of Weddell seals on the shelf. In other parts of their range, Weddell seals mainly consume Antarctic silverfish, with seasonally varying contributions of larger prey such as Antarctic toothfish,

which may increase with seal body size, or age [67]. We do not have age information for the individuals tagged in this study, but it is possible that there is further intraspecific niche separation beyond sexual segregation [68].

A diurnal dive pattern was observed early on in Weddell seals (Ross Sea [26], East Antarctica [28], Weddell Sea [32]), with seals diving deeply in the day and shallowly at night. The shallow depths visited during night-time are now known to overlap with the diel vertical migration of Antarctic silverfish [28,69]. The depths and dive types described in [26,70] correspond closely to our females dive states. In agreement with previous studies, we find that epipelagic (less than 100 m) dives are more likely at night [26,28,32]. The daytime pelagic (less than 250 m) female dives deviate from what we expected for pelagic diving behaviour based on other species (e.g. southern elephant seals [4,7]). It seems likely that at these depths (approx. 200 m) in the Weddell Gyre seals are foraging on some age class of Antarctic silverfish and other notothen species in the day, and moving shallower at night (approx. 50 m).

The biological inferences we have drawn hinge on our analytical approach. The major advantage of HMMs is that we can combine many aspects of diving behaviour and make inferences about possible behavioural classes in multiple dimensions, e.g. depth, duration, hunting behaviour, distance from features, physiological state etc. A drawback of HMMs based on telemetry data collected without direct observations, is that we do not know how well our state interpretations correspond to reality. If we repeated the study, increasing the sample size and measuring physiological parameters (life stage, energetic, and reproductive status), and learning more about the ecosystem and prey field, would provide further valuable information on diving behaviour. The need for information on prey field, prey type, and prey profitability is highlighted by the lack of a clear trend in the proportion of time spent hunting in different dive types.

## 5. Conclusions

In this study, we show that Weddell Sea Weddell seals segregate in horizontal, vertical, and hydrographic space from late summer into midwinter. The residency of males on the continental shelf, and the lack of pelagic diving, suggest that benthic and shallow under-ice habitats provide adequate prey resources to support them. Whether by competition avoidance, or in search of 'better' (i.e. more reliable, abundant,

energetically profitable) foraging opportunities, most of the females in our dataset left the shelf and moved north over the abyssal plain, supported by pelagic and epipelagic resources, before returning east and south towards shelf areas. These movements are associated with access to different hydrographic conditions, which are exploited via the diving patterns we have documented here. This is the earliest evidence both of (1) sex-specific diving patterns in this monomorphic top predator, and of (2) their common reliance on the high-salinity shelf water masses unique to the southern Weddell Sea. Information on intrinsic and external factors, as well as year-round diving behaviour, are needed to begin to understand the mechanisms that give rise to this pattern.

Ethics. Animal capture and handling protocols were reviewed and approved by the University of St Andrews Teaching and Research Ethics Committee (UTREC) and the Animal Welfare and Ethics Committee (AWEC) as part of the ethical review process. Permission to work in the Antarctic was granted under permit no. S7-4/2010 of the Antarctic Act 1994. Capture and deployment of satellite transmitters was carried out by experienced personnel with UK Animal (Scientific Procedures) Act 1986 Personal Licenses.

Data accessibility. The dataset analysed for this article is available in a Zenodo repository at http://doi.org/10.5281/zenodo.3820359 and R code for running the models can be found on GitHub at https://github.com/theoniphotopoulou/emews. An animated gif showing the tracking data through time is also available in a Zenodo repository at https://doi.org/10.5281/zenodo.3985898.

Authors' contributions. T.P. and L.B. collected the data and conceived the study. T.P., J.P., and K.H. developed the methods. T.P. carried out the analysis and wrote the manuscript. K.H. carried out the analysis in electronic supplementary material, S2. J.P. provided high-level statistical support. All authors contributed critically to the manuscript and gave approval for publication.

Competing interests. The authors declare no competing interests.

Funding. TP was supported by a Royal Society Newton International Fellowship (NF170682).

Acknowledgements. Thank you to Marius Årthun and the officers and crew of RRS Ernest Shackleton for their support during cruise ES054. Special thanks to Keith Nicholls for valuable input on Weddell Sea oceanography, and Mike Fedak for productive discussions on pinniped foraging strategies. The data collection was funded by NERCgrant nos NE/G014833/1 and NE/G014086/1. T.P. was supported by a Royal Society Newton International Fellowship (NF170682). K.H. was supported by a Marie-Skłodowska Curie Research Fellowship. The oceanographic data were processed and made freely available by the International MEOP Consortium and the national programs that contribute to it. (http://www.meop.net).

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
