## [Reviewer comments · Proceedings of the Royal Society B: Biological Sciences]

Review History

RSPB-2020-1447.R0 (Original submission)

Review form: Reviewer 1

Recommendation

Major revision is needed (please make suggestions in comments)

Scientific importance: Is the manuscript an original and important contribution to its field?
Excellent

General interest: Is the paper of sufficient general interest?
Good

Quality of the paper: Is the overall quality of the paper suitable?
Good

Is the length of the paper justified?
Yes

Should the paper be seen by a specialist statistical reviewer?
Yes

Do you have any concerns about statistical analyses in this paper? If so, please specify them explicitly in your report.

No

It is a condition of publication that authors make their supporting data, code and materials available - either as supplementary material or hosted in an external repository. Please rate, if applicable, the supporting data on the following criteria.

Is it accessible?

Yes

Is it clear?

Yes

Is it adequate?

Yes

Do you have any ethical concerns with this paper?

No

Comments to the Author

This is a scientifically important manuscript that advances telemetry and analytical approaches by inferring marine predator behaviour using environmental data collected in situ. The authors have provided valuable and interesting insights into Weddell seal behaviour while thoroughly considering oceanography and prey dynamics. While the manuscript is very strong analytically and is generally well-written, I believe the narrative could be substantially improved so that the purpose of the study and main messages are clearer. For example, the title does not reflect the manuscript, as I was expecting to read about underlying drivers of sex differences in behaviour in the Background and Discussion, which was mostly lacking. I therefore suggest either editing the text (particularly the title and conclusion) to better reflect the purpose of the manuscript in characterising dive behaviour using environmental data, or including much more detail on the drivers of sex differences in behaviour in the Background and Discussion. The Abstract also does not clearly state the reason for the study, the study aims or importance of the work. I therefore believe it could be re-written to include these aspects and improve structure. The study aim should also be clearly specified in the introduction (as it is currently at the start of the Discussion). There are several other sentences (indicated below) that would fit in better in other sections of the manuscript to improve flow. I have detailed other suggestions below.

Abstract: The Abstract could be edited to improve structure and better reflect the reason for the study and why the results are important.

Lines 10 – 11. I suggest specifying the four dimensions of ocean habitats or rewording the sentence (as this would be unclear to a reader unfamiliar with the subject area).

Lines 10 – 14. Consider rearranging and editing these sentences. I suggest first introducing the field, then the research gap, then aim of the study followed by the methods used.

Line 13. Suggest stating that the weddell seals are the air-breathing top predator so the reader knows the study species earlier on. Also include the scientific name.

Lines 15 – 17. This sentence seems conflicting as it sounds like the sexes do different things, but the same thing. Please reword to improve clarity e.g. 'Both sexes use high-density continental shelf water masses, but the sexes have different water depth preferences.'

Lines 19 – 20. Suggest stating what the diurnal pattern is.

Lines 20 – 21. This sentence is a bit vague. Suggest removing or adding more detail.

Line 23. Body size is not mentioned in the manuscript. Suggest removing this from the Abstract or discussing hypotheses for sexual segregation (including sexual size dimorphism) in the manuscript.

Background: The Background is interesting and thorough, but there was no information on sex differences in resource use/behaviour. I suggest including at least a paragraph on sex differences in resource use/behaviour, or changing the title of the manuscript to better reflect the purpose of the study. Given the current Background, I believe a more appropriate title would be 'Linking marine predator behaviour with environmental data collected in-situ'.

35 – 37. Please edit this sentence or split it into two sentences to improve clarity.

Lines 42. Suggest inserting 'they are' to improve clarity i.e. 'and they are physiologically more costly'

Lines 40 – 45. Suggest expanding on points in this paragraph.

Lines 52 – 58. 'In this study we characterise ...' These sentences sound like they belong in the last paragraph of the introduction rather than in the middle.

Lines 59 – 63. I think this section about the Weddell Sea could be shortened to include the same information.

Methods: Overall, the methods are very thoroughly explained and the supplementary material expanding on these techniques is useful and relevant. However, the 'Statistical Methods' section would be hard to understand to readers unfamiliar with these techniques.

Line 89. Suggest changing 'behaviour and oceanographic data' to 'movement and oceanographic data' or similar, since behaviour was inferred from the movement data.

Line 91. It would be useful to specify all the variables that were recorded by the CTD-SRDLs.

Figure 1. Please include the sample sizes of males and females in the figure legend.

Line 121. Suggest changing 'using a model-based approach' to 'by fitting a state-space model using the R package foieGras' to include more detail.

Lines 148 – 149. It is unclear why the full model was used and why model selection was not conducted 'based on biological system knowledge', which is also not explained in the supplementary material. Please elaborate.

Line 163. Please explain the 'Viterbi algorithm' and include the reference.

Results: The results are clear and figures nicely presented.

Table 1. Please move the sex column to left side of the table, as this variable is of interest. In the legend, change 'as a the proportion' to 'as the proportion'.

Figure 2. This figure is very useful, although the layout could be improved. It may be clearer to put the graphs for each sex side by side (vertically or horizontally) so that it is easier to compare the density distributions between the sexes. The surface and haul out events could potentially be included as a separate graph.

Lines 187 – 189. Suggest moving these lines on model fit to the Methods section.

Line 192. I would state that females are in the top panels and males in the bottom in the figure legend as opposed to in the text.

Lines 194 – 195. The information about the grey ribbon is repeated in the figure legend. I suggest removing these lines from the text.

Lines 191 – 196. I suggest describing some of the main results from Figure 3 e.g. both sexes had a diurnal pattern in haul out behaviour, with haul out behaviour peaking around 4 – 5pm in all weeks etc.

Lines 201 – 203. These lines could be combined with the above paragraph.

Discussion: The findings are thoroughly discussed, but I believe that the narrative is slightly confused. For example, the first and last paragraphs focus on the importance of sex-specific behaviours, but there is little detail on the drivers of sex-specific behaviours in the rest of the Discussion (this is also lacking in the Background). I suggest including more information on the drivers of sex-specific behaviours or amending the text (particularly in the first paragraph and Conclusion) so that the purpose of the manuscript is focused on linking oceanographic data with behaviour.

Lines 217 – 218. The aim of the study should be moved to the Abstract and Background. I suggest that the first sentence of the Discussion should give the broad finding of the study e.g. 'This study reveals that oceanographic data can be collected by diving marine predators in-situ to characterise their diving behaviour'.

Lines 224 – 226. I recommend moving these lines to the Results section when describing the main results from Figure 3.

Line 232. Suggest adding a sentence to round off the paragraph.

Lines 234 – 235. This question can be removed as the narrative flows without it.

Line 235. Suggest changing ‘the seal’ to ‘seals’.

Line 239. I would personally prefer to read ‘Females may leave the shelf and venture north because...’ rather than asking a question and then answering it.

Line 256 – 257. I would also remove the question.

Line 303. You could add that the proportion of fish in the diet may also increase with body size.

Line 324. It is not clear what the unanswered questions are. Suggest rephrasing to ‘would provide further valuable information on diving behaviour’.

Line 325. Please reword or clarify what you mean by ‘lack of signal’.

Line 330 – 331. Suggest changing (more reliable, abundant, energetically profitable?) to ‘(i.e. more reliable, abundant, and/or energetically profitable)’.

References: Some references have gaps either side of the page numbers that can be removed.

Supplementary Material: The supplementary material is extremely thorough, which would be useful for scientists and statisticians conducting related studies.

S2.3: Suggest amending the layout so that text in the paragraphs is not broken by Figure S2.2.

S4: Please note there is a spelling mistake ‘therefore’.

Review form: Reviewer 2

Recommendation

Accept with minor revision (please list in comments)

Scientific importance: Is the manuscript an original and important contribution to its field?

Excellent

General interest: Is the paper of sufficient general interest?

Excellent

Quality of the paper: Is the overall quality of the paper suitable?

Excellent

Is the length of the paper justified?

Yes

Should the paper be seen by a specialist statistical reviewer?

No

Do you have any concerns about statistical analyses in this paper? If so, please specify them explicitly in your report.

No

It is a condition of publication that authors make their supporting data, code and materials available - either as supplementary material or hosted in an external repository. Please rate, if applicable, the supporting data on the following criteria.

Is it accessible?

Yes

Is it clear?

Yes

Is it adequate?

Yes

Do you have any ethical concerns with this paper?

No

Comments to the Author

This manuscript is a nice piece of work combining a large dataset and a sophisticated statistical approach to study the diving activity of male and female Weddell seal, an ice-obligate to predator, in relation to the physical ocean parameters of its marine habitat. The study was conducted in the southern part of the Weddell Sea where relatively few studies have been conducted so far due to logistical difficulties. A large number of post-moult Weddell seals (10 females, 9 males) were captured on sea-ice from a research vessel and equipped with satellite relayed data loggers recording pressure, temperature and conductivity data transmitted along with the seals positions via the Argos system.

Such large tagging effort to study simultaneously individuals seals of both sexes is remarkable and uncommon. The methodology is well established, and made it possible to collect both behavioural and oceanographic data over more than four months during the austral winter characterized by harsh conditions. Overall, the paper demonstrates clear sex-specific variations in the use of both horizontal and vertical habitat, although males and female Weddell seals are monomorphic, hinting at internal, body mass-independent drivers explaining such differences. The study uses state-of-the art data processing methods to extract relevant information on foraging behaviour from low-resolution data, as well as innovative statistical modelling approach to identify dive states from both diving data, physiography, and a physical ocean parameter (salinity). The paper clearly demonstrates that female Weddell seals use a different strategy from males, by venturing North off the shelf, performing pelagic dives in a relatively warm water mass (MCDW) which are not observed in males which stayed over the shelf diving mostly benthically. Interestingly, females also dived benthically for part of their time and therefore seals of both sexes dived in the cold and dense High Salinity Shelf Water and Ice Shelf Water. By including time of day and season advancement as covariates, the authors demonstrated that females changed their diving patterns over time, with pelagic dives becoming shallower in winter while males did not show clear seasonal trends.

Overall the manuscript is clear and well written. Stronger hypothesis on prey distribution according to vertical structure and water masses could have been made in the background section. In terms of results presentation, it could have been useful to provide a temporal scale for the seal tracks in order to visualise where the seals go over time. It could also have been useful to show basic sea-ice distribution maps, maybe at week 7, 15, and 24, to give an idea of sea-ice conditions during the study. The robust statistical framework supports unequivocal results on the different dives states by integrating three different types of variables (behaviour, physiography, and oceanography) and I found this approach really powerful and innovative. The results obtained are new and clearly contribute to the advancement of our understanding of the foraging ecology of a key top predator of the sea-ice zone. The discussion is interesting with new ideas on environmental predictability and movement ecology of the seals. I think the authors also should consider the role of diurnal/seasonal changes of light ability when discussing circadian and seasonal variations in diving behaviour. I strongly recommend the publication of the manuscript after the minor revisions suggested above.

Decision letter (RSPB-2020-1447.R0)

29-Jul-2020

Dear Dr Photopoulou:

Your manuscript has now been peer reviewed and the reviews have been assessed by an Associate Editor. The reviewers' comments (not including confidential comments to the Editor) and the comments from the Associate Editor are included at the end of this email for your reference. As you will see, the reviewers and the Editors have raised some concerns with your manuscript and we would like to invite you to revise your manuscript to address them.

Research ethics:

Use of animals and field studies:

It is a condition of publication that you make available the data and research materials supporting the results in the article. Please see our Data Sharing Policies (<https://royalsociety.org/journals/authors/author-guidelines/#data>). Datasets should be deposited in an appropriate publicly available repository and details of the associated accession number, link or DOI to the datasets must be included in the Data Accessibility section of the

article (<https://royalsociety.org/journals/ethics-policies/data-sharing-mining/>). Reference(s) to datasets should also be included in the reference list of the article with DOIs (where available).

Please submit a copy of your revised paper within three weeks. If we do not hear from you within this time your manuscript will be rejected. If you are unable to meet this deadline please let us know as soon as possible, as we may be able to grant a short extension.

Best wishes,
Dr Daniel Costa
mailto: proceedingsb@royalsociety.org

Associate Editor

Board Member: 1

Comments to Author:

two reviews have been received, both positive, with one including a range of detailed suggestions for improving the clarity and flow in parts of the ms. In particular they ask for clearer explanation of the statistical methods used, which I have sympathy with, it is important that the wider readership can access clear understanding of the approaches applied, and not only those already specialist in the field. I would recommend careful revision in the light of the detailed comments.

Reviewer(s)' Comments to Author:

Referee: 1

Comments to the Author(s)

This is a scientifically important manuscript that advances telemetry and analytical approaches by inferring marine predator behaviour using environmental data collected in situ. The authors

have provided valuable and interesting insights into Weddell seal behaviour while thoroughly considering oceanography and prey dynamics. While the manuscript is very strong analytically and is generally well-written, I believe the narrative could be substantially improved so that the purpose of the study and main messages are clearer. For example, the title does not reflect the manuscript, as I was expecting to read about underlying drivers of sex differences in behaviour in the Background and Discussion, which was mostly lacking. I therefore suggest either editing the text (particularly the title and conclusion) to better reflect the purpose of the manuscript in characterising dive behaviour using environmental data, or including much more detail on the drivers of sex differences in behaviour in the Background and Discussion. The Abstract also does not clearly state the reason for the study, the study aims or importance of the work. I therefore believe it could be re-written to include these aspects and improve structure. The study aim should also be clearly specified in the introduction (as it is currently at the start of the Discussion). There are several other sentences (indicated below) that would fit in better in other sections of the manuscript to improve flow. I have detailed other suggestions below.

Abstract: The Abstract could be edited to improve structure and better reflect the reason for the study and why the results are important.

Lines 10 – 11. I suggest specifying the four dimensions of ocean habitats or rewording the sentence (as this would be unclear to a reader unfamiliar with the subject area).

Lines 10 – 14. Consider rearranging and editing these sentences. I suggest first introducing the field, then the research gap, then aim of the study followed by the methods used.

Line 13. Suggest stating that the weddell seals are the air-breathing top predator so the reader knows the study species earlier on. Also include the scientific name.

Lines 15 – 17. This sentence seems conflicting as it sounds like the sexes do different things, but the same thing. Please reword to improve clarity e.g. 'Both sexes use high-density continental shelf water masses, but the sexes have different water depth preferences.'

Lines 19 – 20. Suggest stating what the diurnal pattern is.

Lines 20 – 21. This sentence is a bit vague. Suggest removing or adding more detail.

Line 23. Body size is not mentioned in the manuscript. Suggest removing this from the Abstract or discussing hypotheses for sexual segregation (including sexual size dimorphism) in the manuscript.

Background: The Background is interesting and thorough, but there was no information on sex differences in resource use/behaviour. I suggest including at least a paragraph on sex differences in resource use/behaviour, or changing the title of the manuscript to better reflect the purpose of the study. Given the current Background, I believe a more appropriate title would be 'Linking marine predator behaviour with environmental data collected in-situ'.

35 – 37. Please edit this sentence or split it into two sentences to improve clarity.

Lines 42. Suggest inserting 'they are' to improve clarity i.e. 'and they are physiologically more costly'

Lines 40 – 45. Suggest expanding on points in this paragraph.

Lines 52 – 58. 'In this study we characterise ...' These sentences sound like they belong in the last paragraph of the introduction rather than in the middle.

Lines 59 – 63. I think this section about the Weddell Sea could be shortened to include the same information.

Methods: Overall, the methods are very thoroughly explained and the supplementary material expanding on these techniques is useful and relevant. However, the 'Statistical Methods' section would be hard to understand to readers unfamiliar with these techniques.

Line 89. Suggest changing 'behaviour and oceanographic data' to 'movement and oceanographic data' or similar, since behaviour was inferred from the movement data.

Line 91. It would be useful to specify all the variables that were recorded by the CTD-SRDLs.

Figure 1. Please include the sample sizes of males and females in the figure legend.

Line 121. Suggest changing 'using a model-based approach' to 'by fitting a state-space model using the R package foieGras' to include more detail.

Lines 148 – 149. It is unclear why the full model was used and why model selection was not conducted ‘based on biological system knowledge’, which is also not explained in the supplementary material. Please elaborate.

Line 163. Please explain the ‘Viterbi algorithm’ and include the reference.

Results: The results are clear and figures nicely presented.

Table 1. Please move the sex column to left side of the table, as this variable is of interest. In the legend, change ‘as a the proportion’ to ‘as the proportion’.

Figure 2. This figure is very useful, although the layout could be improved. It may be clearer to put the graphs for each sex side by side (vertically or horizontally) so that it is easier to compare the density distributions between the sexes. The surface and haul out events could potentially be included as a separate graph.

Lines 187 – 189. Suggest moving these lines on model fit to the Methods section.

Line 192. I would state that females are in the top panels and males in the bottom in the figure legend as opposed to in the text.

Lines 194 – 195. The information about the grey ribbon is repeated in the figure legend. I suggest removing these lines from the text.

Lines 191 – 196. I suggest describing some of the main results from Figure 3 e.g. both sexes had a diurnal pattern in haul out behaviour, with haul out behaviour peaking around 4 – 5pm in all weeks etc.

Lines 201 – 203. These lines could be combined with the above paragraph.

Discussion: The findings are thoroughly discussed, but I believe that the narrative is slightly confused. For example, the first and last paragraphs focus on the importance of sex-specific behaviours, but there is little detail on the drivers of sex-specific behaviours in the rest of the Discussion (this is also lacking in the Background). I suggest including more information on the drivers of sex-specific behaviours or amending the text (particularly in the first paragraph and Conclusion) so that the purpose of the manuscript is focused on linking oceanographic data with behaviour.

Lines 217 – 218. The aim of the study should be moved to the Abstract and Background. I suggest that the first sentence of the Discussion should give the broad finding of the study e.g. ‘This study reveals that oceanographic data can be collected by diving marine predators in-situ to characterise their diving behaviour’.

Lines 224 – 226. I recommend moving these lines to the Results section when describing the main results from Figure 3.

Line 232. Suggest adding a sentence to round off the paragraph.

Lines 234 – 235. This question can be removed as the narrative flows without it.

Line 235. Suggest changing ‘the seal’ to ‘seals’.

Line 239. I would personally prefer to read ‘Females may leave the shelf and venture north because...’ rather than asking a question and then answering it.

Line 256 – 257. I would also remove the question.

Line 303. You could add that the proportion of fish in the diet may also increase with body size.

Line 324. It is not clear what the unanswered questions are. Suggest rephrasing to ‘would provide further valuable information on diving behaviour’.

Line 325. Please reword or clarify what you mean by ‘lack of signal’.

Line 330 – 331. Suggest changing (more reliable, abundant, energetically profitable?) to ‘(i.e. more reliable, abundant, and/or energetically profitable)’.

References: Some references have gaps either side of the page numbers that can be removed.

Supplementary Material: The supplementary material is extremely thorough, which would be useful for scientists and statisticians conducting related studies.

S2.3: Suggest amending the layout so that text in the paragraphs is not broken by Figure S2.2.

S4: Please note there is a spelling mistake ‘therefore’.

Referee: 2

Comments to the Author(s)

This manuscript is a nice piece of work combining a large dataset and a sophisticated statistical approach to study the diving activity of male and female Weddell seal, an ice-obligate to predator, in relation to the physical ocean parameters of its marine habitat. The study was conducted in the southern part of the Weddell Sea where relatively few studies have been conducted so far due to logistical difficulties. A large number of post-moult Weddell seals (10 females, 9 males) were captured on sea-ice from a research vessel and equipped with satellite relayed data loggers recording pressure, temperature and conductivity data transmitted along with the seals positions via the Argos system.

Such large tagging effort to study simultaneously individuals seals of both sexes is remarkable and uncommon. The methodology is well established, and made it possible to collect both behavioural and oceanographic data over more than four months during the austral winter characterized by harsh conditions. Overall, the paper demonstrates clear sex-specific variations in the use of both horizontal and vertical habitat, although males and female Weddell seals are monomorphic, hinting at internal, body mass-independent drivers explaining such differences. The study uses state-of-the art data processing methods to extract relevant information on foraging behaviour from low-resolution data, as well as innovative statistical modelling approach to identify dive states from both diving data, physiography, and a physical ocean parameter (salinity). The paper clearly demonstrates that female Weddell seals use a different strategy from males, by venturing North off the shelf, performing pelagic dives in a relatively warm water mass (MCDW) which are not observed in males which stayed over the shelf diving mostly benthically. Interestingly, females also dived benthically for part of their time and therefore seals of both sexes dived in the cold and dense High Salinity Shelf Water and Ice Shelf Water. By including time of day and season advancement as covariates, the authors demonstrated that females changed their diving patterns over time, with pelagic dives becoming shallower in winter while males did not show clear seasonal trends.

Overall the manuscript is clear and well written. Stronger hypothesis on prey distribution according to vertical structure and water masses could have been made in the background section. In terms of results presentation, it could have been useful to provide a temporal scale for the seal tracks in order to visualise where the seals go over time. It could also have been useful to show basic sea-ice distribution maps, maybe at week 7, 15, and 24, to give an idea of sea-ice conditions during the study. The robust statistical framework supports unequivocal results on the different dives states by integrating three different types of variables (behaviour, physiography, and oceanography) and I found this approach really powerful and innovative.

The results obtained are new and clearly contribute to the advancement of our understanding of the foraging ecology of a key top predator of the sea-ice zone. The discussion is interesting with new ideas on environmental predictability and movement ecology of the seals. I think the authors also should consider the role of diurnal/seasonal changes of light ability when discussing circadian and seasonal variations in diving behaviour. I strongly recommend the publication of the manuscript after the minor revisions suggested above.

Author's Response to Decision Letter for (RSPB-2020-1447.R0)

See Appendix A.

RSPB-2020-1447.R1 (Revision)

Review form: Reviewer 1

Recommendation

Accept as is

Scientific importance: Is the manuscript an original and important contribution to its field?

Excellent

General interest: Is the paper of sufficient general interest?

Excellent

Quality of the paper: Is the overall quality of the paper suitable?

Excellent

Is the length of the paper justified?

Yes

Should the paper be seen by a specialist statistical reviewer?

No

Do you have any concerns about statistical analyses in this paper? If so, please specify them explicitly in your report.

No

It is a condition of publication that authors make their supporting data, code and materials available - either as supplementary material or hosted in an external repository. Please rate, if applicable, the supporting data on the following criteria.

Is it accessible?

Yes

Is it clear?

Yes

Is it adequate?

Yes

Do you have any ethical concerns with this paper?

No

Comments to the Author

The authors have replied to my queries. Note that providing sea-ice concentration instead of ice extent would be much more informative, but this is up to the authors, otherwise the paper is ready to be published.

Review form: Reviewer 2

Recommendation

Accept as is

Scientific importance: Is the manuscript an original and important contribution to its field?
Excellent

General interest: Is the paper of sufficient general interest?
Good

Quality of the paper: Is the overall quality of the paper suitable?
Excellent

Is the length of the paper justified?
Yes

Should the paper be seen by a specialist statistical reviewer?
Yes

Do you have any concerns about statistical analyses in this paper? If so, please specify them explicitly in your report.
No

It is a condition of publication that authors make their supporting data, code and materials available - either as supplementary material or hosted in an external repository. Please rate, if applicable, the supporting data on the following criteria.

Is it accessible?
Yes

Is it clear?
Yes

Is it adequate?
Yes

Do you have any ethical concerns with this paper?
No

Comments to the Author

The authors have done an excellent job at addressing the reviewers' feedback. I believe the narrative has substantially improved and the manuscript is easier to read. The animated tracks are also a nice addition. I noticed that a sentence is repeated (lines 295 - 296) that should be deleted, but I have no other suggestions. I look forward to seeing this manuscript in its published form and I congratulate the authors.

Decision letter (RSPB-2020-1447.R1)

21-Sep-2020

Dear Dr Photopoulou

I am pleased to inform you that your manuscript entitled "Sex-specific variation in the use of vertical habitat by a resident Antarctic top predator" has been accepted for publication in Proceedings B.

Open Access

Paper charges

Sincerely,

Dr Daniel Costa

Associate Editor:

Comments to Author:

Both reviewers are happy with the revisions, and I would also thank the authors for the very thorough work in detailing their responses and changes made. Note that a sentence at l295 is repeated and this should be corrected prior to proofing.

Appendix A

Dr Theoni Photopoulou
Scottish Oceans Institute
School of Biology
University of St Andrews
St Andrews, Scotland

20 August 2020

Dear Dr Costa,
Academic Editor, Proceedings of the Royal Society B

REVISED SUBMISSION OF MANUSCRIPT RSPB-2020-1447
*Sex-specific variation in the use of vertical habitat by a resident
Antarctic top predator*

Many thanks to the two anonymous reviewers for constructive comments on our manuscript submitted to Proceedings of the Royal Society B.

Both reviewers make suggestions for changes or improvements to the text. We have followed their suggestions and added details in response to their comments. The changes made can be seen in the table below where we have addressed the reviewers' comments point-by-point. We underline actions taken in response to each comment.

While we have included additional information wherever it was requested by the reviewers, we were limited by the maximum length of the manuscript, so we have had to be succinct. We have made small changes to the wording throughout the manuscript to further reduce the word count, accommodate the additional information requested by the reviewers, and keep it within the journal's page limit. All of our line number references below correspond to the revised manuscript with tracked changes, included in this document below our responses.

We have addressed the reviewers' comments as best as we can and hope you will find our responses satisfactory. We look forward to hearing from you and hope that you find our revised manuscript suitable for publication in Proceedings of the Royal Society B.

Sincerely,

Dr Theoni Photopoulou
on behalf of all the authors

Responses to Reviewer 1

Comment	Response
While the manuscript is very strong analytically and is generally well-written, I believe the narrative could be substantially improved so that the purpose of the study and main messages are clearer. For example, the title does not reflect the manuscript, as I was expecting to read about underlying drivers of sex differences in behaviour in the Background and Discussion, which was mostly lacking. I therefore suggest either editing the text (particularly the title and conclusion) to better reflect the purpose of the manuscript in characterising dive behaviour using environmental data, or including much more detail on the drivers of sex differences in behaviour in the Background and Discussion.	Thank you for pointing this out. To improve the narrative, we have added detail on the drivers of sex differences in behaviour to the Background (lines 85-91), Discussion (lines 255-257, 281-284, 292-298) and Conclusion (lines 389-390) sections and kept the original title.
The Abstract also does not clearly state the reason for the study, the study aims or importance of the work. I therefore believe it could be re-written to include these aspects and improve structure. The study aim should also be clearly specified in the introduction (as it is currently at the start of the Discussion). There are several other sentences (indicated below) that would fit in better in other sections of the manuscript to improve flow. I have detailed other suggestions below.	Thank you for these suggestions, we have revised the abstract to include this information and followed the other suggestions you have outlined. We now state the motivation for the study on line 14 “This dimension of space use is not captured if we only consider horizontal movement”; the aim of the study on line 15 “To identify different diving behaviours and understand usage patterns of vertically distributed habitat...” and the implications of the results on line 26 “The differences in habitat use in a resident, sexually monomorphic Antarctic top predator suggest a different set of needs and constraints operating at the intraspecific level, which are not driven by body size.”
Abstract: The Abstract could be edited to improve structure and	We have revised the Abstract to clarify the motivation for the study, its aims and

better reflect the reason for the study and why the results are important.

the significance of the main results. We include details and line numbers in our response to the previous comment, above.

Lines 10 – 11. I suggest specifying the four dimensions of ocean habitats or rewording the sentence (as this would be unclear to a reader unfamiliar with the subject area).

Included on line 11 of the revised manuscript: (space, depth, time) following the European Marine Board document Navigating the Future IV on a four-dimensional ocean (<https://bit.ly/3kH6d2n>)

Lines 10 – 14. Consider rearranging and editing these sentences. I suggest first introducing the field, then the research gap, then aim of the study followed by the methods used.

Done as suggested: see revised text on lines 11-17.

Line 13. Suggest stating that the weddell seals are the air-breathing top predator so the reader knows the study species earlier on. Also include the scientific name.

Text rearranged to mention Weddell seals right after “air-breathing top predator” on line 16.

Lines 15 – 17. This sentence seems conflicting as it sounds like the sexes do different things, but the same thing. Please reword to improve clarity e.g. ‘Both sexes use high-density continental shelf water masses, but the sexes have different water depth preferences.’

Thanks for this, we wanted to put across that there is some overlap in behaviour but that female seals also do something that males do not. We have revised this sentence, now on lines 18-21.

Lines 19 – 20. Suggest stating what the diurnal pattern is.

We have included this in brackets on line 24.

Lines 20 – 21. This sentence is a bit vague. Suggest removing or adding more detail.

We have removed this sentence.

Line 23. Body size is not mentioned in the manuscript. Suggest removing this from the Abstract or discussing hypotheses for sexual segregation (including sexual size dimorphism) in the manuscript.

We have kept this sentence as is in the Abstract and included additional information on sexual segregation to the Background and Discussion sections (lines 85-91, 255-257, 281-284, 292-289, 389-390).

Background: The Background is interesting and thorough, but there was no information on sex differences in resource

This is a useful point, thank you. We feel that the sex-specific differences presented in this manuscript are one of the most exciting and novel findings and

use/behaviour. I suggest including at least a paragraph on sex differences in resource use/behaviour, or changing the title of the manuscript to better reflect the purpose of the study. Given the current Background, I believe a more appropriate title would be 'Linking marine predator behaviour with environmental data collected in-situ'.

we want to keep that in the title. To make the title more representative of the narrative, we have included information on sex differences in resource use and behaviour in the Background, Discussion and Conclusion (lines 85-91, 255-257, 281-284, 292-289, 389-390).

35 – 37. Please edit this sentence or split it into two sentences to improve clarity.

We have revised this sentence for clarity, lines 41-42.

Lines 42. Suggest inserting 'they are' to improve clarity i.e. 'and they are physiologically more costly'

Inserted, line 47.

Lines 40 – 45. Suggest expanding on points in this paragraph.

We have added a sentence to clarify the main point being made (line 49-50). however, having added a paragraph on drivers in sex differences, we could not add more text here to keep within the 10-page limit of the journal. We would be very happy to expand it further if the page limit allows.

Lines 52 – 58. 'In this study we characterise ...' These sentences sound like they belong in the last paragraph of the introduction rather than in the middle.

We have removed these sentences and revised the paragraph for clarity, lines 62-65.

Lines 59 – 63. I think this section about the Weddell Sea could be shortened to include the same information.

We have shortened this section to a sentence, lines 70-72.

Methods: Overall, the methods are very thoroughly explained and the supplementary material expanding on these techniques is useful and relevant. However, the 'Statistical Methods' section would be hard to understand to readers unfamiliar with these techniques.

We have revised the Statistical Methods section to improve clarity and include more detail that would help an unfamiliar reader understand the approach, lines 143-147, 156-161.

Line 89. Suggest changing 'behaviour and oceanographic data' to 'movement and oceanographic data' or similar, since behaviour

Done, line 108.

was inferred from the movement data.

Line 91. It would be useful to specify all the variables that were recorded by the CTD-SRDLs.

Done, line 108.

Figure 1. Please include the sample sizes of males and females in the figure legend.

Done, Figure 1 caption.

Line 121. Suggest changing ‘using a model-based approach’ to ‘by fitting a state-space model using the R package foieGras’ to include more detail.

Done, line 140-141.

Lines 148 – 149. It is unclear why the full model was used and why model selection was not conducted ‘based on biological system knowledge’, which is also not explained in the supplementary material. Please elaborate.

We appreciate that this was a bit unclear. The reason for including covariates in the model was to explore temporal effects on diving behaviour, so we only considered temporal covariates. We explored using light level (lux) instead of time of day as a fine-scale temporal covariate, but the pattern was less clear with this formulation.

Histograms of the raw data showed clear diurnal and clear seasonal patterns so it did not make sense to include one without the other. In addition, there is an extreme change in environmental conditions in the Antarctic over the course of our study period, which spans the spring solstice and midwinter. This made it important to include an interaction between time of day and week of the year, to allow for the fact that time of day may have a different effect on the probability of transitioning between states at different stages in the seasonal progression.

We have added details about why we chose not to do model selection to Supplementary Material S4.1.3, and we refer the reader to it in this section, lines

	177-178.
Line 163. Please explain the ‘Viterbi algorithm’ and include the reference.	The Viterbi algorithm is the standard algorithm to decode the unobserved states by calculating the most likely state sequence under a given HMM. It is a standard part of interpreting the model results and that is why we have not justified our use of it. However, we have added a section to Supplementary Material S4 briefly explaining what it does, and we have added a reference to the main text (lines 192-193).
Results: The results are clear and figures nicely presented.	Thank you! A lot of thought went into this.
Table 1. Please move the sex column to left side of the table, as this variable is of interest. In the legend, change ‘as a the proportion’ to ‘as the proportion’.	Done, please see the revised Table 1 and legend.
Figure 2. This figure is very useful, although the layout could be improved. It may be clearer to put the graphs for each sex side by side (vertically or horizontally) so that it is easier to compare the density distributions between the sexes. The surface and haul out events could potentially be included as a separate graph.	We have revised this figure and its legend following your suggestions. We have kept the surface and haulout densities in the plot, but we have made them a little smaller so that they are not the main focus of the figure. Please see revised Figure 2, where we have stacked the plots vertically to allow for easy comparisons between the distributions.
Lines 187 – 189. Suggest moving these lines on model fit to the Methods section.	We have moved these lines to the end of the Methods section, lines 193-196.
Line 192. I would state that females are in the top panels and males in the bottom in the figure legend as opposed to in the text.	Done, line 226 and revised Figure 3.
Lines 194 – 195. The information about the grey ribbon is repeated in the figure legend. I suggest removing these lines from the text.	Done, lines 228-229.
Lines 191 – 196. I suggest describing some of the main results from Figure 3 e.g. both sexes had a	We have added some results here, lines 229-233.

diurnal pattern in haul out behaviour, with haul out behaviour peaking around 4 – 5pm in all weeks etc.

Lines 201 – 203. These lines could be combined with the above paragraph.

Discussion: The findings are thoroughly discussed, but I believe that the narrative is slightly confused. For example, the first and last paragraphs focus on the importance of sex-specific behaviours, but there is little detail on the drivers of sex-specific behaviours in the rest of the Discussion (this is also lacking in the Background). I suggest including more information on the drivers of sex-specific behaviours or amending the text (particularly in the first paragraph and Conclusion) so that the purpose of the manuscript is focused on linking oceanographic data with behaviour.

Lines 217 – 218. The aim of the study should be moved to the Abstract and Background. I suggest that the first sentence of the Discussion should give the broad finding of the study e.g. ‘This study reveals that oceanographic data can be collected by diving marine predators in-situ to characterise their diving behaviour’.

Lines 224 – 226. I recommend moving these lines to the Results section when describing the main results from Figure 3.

Line 232. Suggest adding a sentence to round off the paragraph.

Done, lines 238.

Thanks for this. We have revised the Discussion to start with the main finding of the study (lines 255-257) and added details about how our results fit in with various hypotheses for sex-specific differences in habitat segregation both in the Background (lines 85-91), Discussion (lines 281-284, 292-289) and Conclusion (389-390).

We have included the aim in the Abstract (lines 15-16) and the Background (lines 62-65 and reiterated lines 100-104).

We have also amended the first sentence of the Discussion, lines 255-257: “We show that oceanographic data collected *in situ* by diving marine predators can be used to characterise their preferences for vertically distributed habitat and reveal intraspecific variability...”

We have removed these lines, as suggested, lines (264-266).

We have added a sentence to round off, lines 272-273: “suggesting that male diving behaviour does not track seasonally varying limitations or

	opportunities”.
Lines 234 – 235. This question can be removed as the narrative flows without it.	We have removed this question, lines 275-276.
Line 235. Suggest changing ‘the seal’ to ‘seals’.	Done, line 276.
Line 239. I would personally prefer to read ‘Females may leave the shelf and venture north because...’ rather than asking a question and then answering it.	We have revised this sentence to remove the question, lines 280-284.
Line 256 – 257. I would also remove the question.	We have removed this question, lines 305-306.
Line 303. You could add that the proportion of fish in the diet may also increase with body size.	We have included that large fish may contribute a greater proportion to the diet as seals get older, line 352-353.
Line 324. It is not clear what the unanswered questions are. Suggest rephrasing to ‘would provide further valuable information on diving behaviour’.	We have revised this sentence to include your suggestion, lines 375-376.
Line 325. Please reword or clarify what you mean by ‘lack of signal’.	We have replaced “signal” with “a clear trend”, line 377.
Line 330 – 331. Suggest changing (more reliable, abundant, energetically profitable?)’ to ‘(i.e. more reliable, abundant, and/or energetically profitable)’.	Revised as suggested, lines 382-383.
References: Some references have gaps either side of the page numbers that can be removed.	Done , thanks for noticing that.
Supplementary Material: The supplementary material is extremely thorough, which would be useful for scientists and statisticians conducting related studies.	Thank you, that was exactly our intention.
S2.3: Suggest amending the layout so that text in the paragraphs is not broken by Figure S2.2.	We have amended the layout as suggested. Please see revised Supplementary Material S2.
S4: Please note there is a spelling mistake ‘therefore’.	Corrected, thank you.

Responses to Reviewer 2

Comment	Response
Stronger hypothesis on prey distribution according to vertical structure and water masses could have been made in the background section.	We have revised the wording (lines 39-42) and added a sentence (line 49-50) to strengthen this hypothesis.
In terms of results presentation, it could have been useful to provide a temporal scale for the seal tracks in order to visualise where the seals go over time.	We have created an animation of the tracks on a background map showing the edge of the Antarctic ice sheet. Unfortunately, the background image does not change with the seal tracks to provide a picture of evolving sea ice concentration, but we also provide static images of contemporary seal tracks and sea ice, in response to the next comment. The animation is open access and can be downloaded as a .gif from Zenodo at the following URL https://doi.org/10.5281/zenodo.3985898 We have added this URL to the Availability of data and material section
It could also have been useful to show basic sea-ice distribution maps, maybe at week 7, 15, and 24, to give an idea of sea-ice conditions during the study.	We have added a figure to Supplementary Material S1, Figure S1.2, showing the monthly ice extent and associated seal tracks (colour coded by sex as in Figure 1 of the main text) for months February, April and June of 2011.
I think the authors also should consider the role of diurnal/seasonal changes of light ability when discussing circadian and seasonal variations in diving behaviour.	We explored the role of light conditions during model fitting and found that, although there is a pattern which corresponds with what we found for time of day, it is less clear. The time of day effect also shows that the circadian pattern persists, though less strongly, even after there is very little variation in day-night light conditions. We have added a section to Supplementary Material S4.1.3 explaining the model selection rationale, where we also discuss this.